

**Intra-annual variations of regional aerosol optical depth, vertical distribution, and**
**particle types from multiple satellite and ground-based observational datasets**
Bin Zhao[1], Jonathan H. Jiang[2], David J. Diner[2], Hui Su[2], Yu Gu[1], Kuo-Nan Liou[1], Zhe Jiang[1],
Lei Huang[1], Yoshi Takano[1], Xuehua Fan[1], and Ali H. Omar[3]
[1]Joint Institute for Regional Earth System Science and Engineering and Department of
Atmospheric and Oceanic Sciences, University of California, Los Angeles, California, USA.
[2]Jet propulsion Laboratory, California Institute of Technology, Pasadena, California, USA.
[3]NASA Langley Research Center, Hampton, Virginia, USA.
Corresponding author: Bin Zhao (zhaob1206@ucla.edu)



## Abstract

The relatively short lifetimes of aerosols in the atmosphere result in climatic and health effects that are strongly dependent on intra-annual variations in particle concentrations. While many studies have examined the seasonal and diurnal variations of regional aerosol optical depth (AOD), understanding the temporal variations in aerosol vertical distribution and particle types is also important for accurate computation of aerosol radiative effects. In this paper, we combine the observations from four satellite-borne sensors and ground-based AOD and fine particle ($PM_{2.5}$) measurements to investigate the seasonal and diurnal variations of aerosol column loading, vertical distribution, and particle types over three populous regions: the Eastern United States (EUS), Western Europe (WEU), and Eastern and Central China (ECC). In all three regions, column AOD, as well as AOD higher than 800 m above ground level, peaks in summer/spring probably due to accelerated formation of secondary aerosols and hygroscopic growth. However, AOD at height below 800 m mostly peaks in winter except that a second maximum in summer occurs over the EUS region, which is consistent with observed temporal trends in surface $PM_{2.5}$ concentrations. AOD due to fine particles (< 0.7 μm diameter) is much larger in spring/summer than in winter over all three regions, whereas coarse mode AOD (> 1.4 μm diameter) generally shows less variability, except for the ECC region where a peak occurs in spring, consistent with the prevalence of airborne dust during this season. When aerosols are classified according to sources, the dominant type is associated with anthropogenic air pollution, which has a similar seasonal pattern as total AOD. Dust and sea-spray aerosols in the WEU region peak in summer and winter, respectively, but do not show an obvious seasonal pattern in the EUS region. Smoke aerosols, as well as absorbing aerosols, present an obvious unimodal distribution with a maximum occurring in summer over the EUS and WEU regions, whereas they follow a bimodal



distribution with peaks in August and March (due to crop residue burning) over the ECC region. In general, the nighttime-daytime AOD difference is more positive in summer than in winter, likely attributable to a larger diurnal temperature range in summer. Smoke AOD is much higher in the nighttime than in the daytime. The results of this study can help to improve the current estimates of the climatic and health impacts of aerosols.

## 1 Introduction

Aerosols have adverse effects on human health (Lelieveld et al., 2015) and play a key role in Earth's climate through aerosol-radiation interactions (McCormick and Ludwig, 1967) and aerosol-cloud interactions (Twomey, 1977; Albrecht, 1989; Garrett and Zhao, 2006). Compared with long-lived climate forcers such as $CO_2$, aerosols have relatively short lifetimes and hence large spatiotemporal variability (Unger et al., 2008; Shindell et al., 2009). While the climatic effects of $CO_2$ are mainly induced by inter-annual concentration changes, the climatic and health effects of aerosols also strongly depend on their intra-annual (seasonal and diurnal) variability.

Aerosol optical depth (AOD) has been widely used to represent the column aerosol loading and to assess the aerosol impacts on radiation, clouds, and precipitation (Ma et al., 2014; Niu and Li, 2012; Zhao et al., 2018; Song et al., 2017). However, the wide ranges of particle scattering and absorption properties mean that even for the same AOD, different aerosol components have different effects on not only the magnitude, but also the sign, of aerosol radiative forcing (IPCC, 2013; Gu et al., 2006). IPCC (2013) estimates that the historical global mean direct radiative forcings due to sulfate, organic carbon (OC), black carbon (BC), and mineral dust are −0.40, −0.19, +0.36, and −0.10 W m$^{-2}$, respectively. Furthermore, absorbing

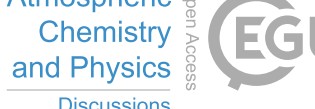

and non-absorbing aerosols have been found to have very different impacts on the development
of convective clouds (Massie et al., 2016; Ramanathan et al., 2005; Rosenfeld et al., 2008).
Besides aerosol type, perturbation of aerosol vertical distribution influences the vertical profile
of heating rate (Johnson et al., 2008; Guan et al., 2010; Zhang et al., 2013), which subsequently
modifies the atmospheric stability and convective strength (Ramanathan et al., 2007), with
potential changes in regional circulation (Ramanathan et al., 2001) and cloud cover (Johnson et
al., 2004). Understanding aerosol variability as a function of height is also important because the
health impacts of aerosols are only associated with those present near the surface, where they are
inhaled. For these reasons, systematic analyses of the intra-annual variations of aerosol vertical
distribution and particle types, in addition to total column AOD, are necessary to improve our
understanding of aerosol climatic and health effects.

Numerous studies have investigated the seasonal variations of AOD at global and

regional scales using satellite observations (e.g., Kim et al., 2007; Song et al., 2009; Mehta et al.,
2016; Mao et al., 2014). By comparison, most previous studies of the temporal variations of
aerosol vertical distributions and aerosol types have been confined to only a few sites due to
coverage limitations associated with reliance on ground-based instruments (e.g., Liu et al., 2012;
Matthias et al., 2004). Despite continuous advancement of remote sensing technology and
emergence of new spaceborne sensors, only limited number of studies have utilized satellite
observations to examine the seasonal and/or diurnal variations of aerosol profiles and/or types at
regional or larger scales (Huang et al., 2013; Kahn and Gaitley, 2015; Yu et al., 2010; Li et al.,
2016). Huang et al. (2013) analyzed the seasonal and diurnal variations of aerosol extinction
profile and type distribution using 5-year observations from the Cloud-Aerosol Lidar and
Infrared Pathfinder Satellite Observations (CALIPSO). Kahn and Gaitley (2015) examined the





spatiotemporal variations of aerosol types retrieved by the Multi-angle Imaging
SpectroRadiometer (MISR). Different satellite-borne sensors, such as MISR, CALIPSO, and
Moderate resolution Imaging Spectroradiometer (MODIS), employ different principles of
measurement and retrieval, and therefore provide different sensitivities to column AOD, aerosol
types, and vertical profiles. Therefore, integration of data from multiple satellites and ground-
based observational networks makes it possible to deepen our understanding of the intra-annual
variations of aerosol loadings, profiles, and types.
In this study, we investigate the seasonal and diurnal variations of aerosol column loading,
vertical distribution, and particle types using multiple satellite and ground-based observational
datasets covering the period from 2007 to 2016. The purpose is to assess the consistency among
various datasets and provide a comprehensive characterization of aerosol properties in polluted
regions to facilitate future studies of aerosol climate effects and local air quality issues. The data
are from MISR, MODIS, CALIPSO, Aerosol Robotic Network (AERONET), and surface $PM_{2.5}$
monitors. Consistent with our previous study (Zhao et al., 2017), we selected three populous
regions which have experienced substantial anthropogenic pollution (Wang et al., 2017; Wang et
al., 2014) and have received considerable attention in other climate studies: the Eastern United
States (EUS; 29º-45º N, 70º-98º W), Western Europe (WEU; 37º-59º N, 10º W-17º E), and
Eastern and Central China (ECC; 21º-41º N, 102º-122º E). The geographical boundaries of these
regions are shown in Fig. 1.

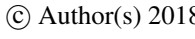



## 2 Data and Methods

2.1 Satellite data

We obtain retrievals of total column AOD as well as AOD for various height ranges and aerosol types from MISR (flying on the Terra satellite), MODIS (Terra and Aqua), and the Cloud-Aerosol Lidar with Orthogonal Polarization (CALIOP) on CALIPSO.

MISR observes the Earth with moderately high spatial resolution (275 m to 1.1 km) at 9 along-track viewing angles in each of 4 visible/near-infrared spectral bands, which enables the partitioning of AOD by particle type over both land and ocean, in addition to retrieval of total AOD (Kahn and Gaitley, 2015; Kahn et al., 2001). Its observations provide near-global coverage every 9 days (Diner et al., 1998). We make use of the Level 3 monthly global aerosol product (MIL3MAE) version F15_0031, which is generated at a spatial resolution of $0.5° \times 0.5°$. The variables used in the analysis are total AOD at 555 nm as well as AODs for six aerosol components, namely small (< 0.7 μm diameter), medium (0.7-1.4 μm diameter), large (> 1.4 μm diameter), spherical, non-spherical, and absorbing. Based on comparison with ground-based AERONET measurements, the errors in MISR AOD data are on the order of $\pm 0.05$ or $\pm(0.20 \times$ AOD), whichever is larger (Kahn et al., 2005; Kahn et al., 2010). In addition, retrieval of MISR particle property information from individual retrievals is considered to be reliable when AOD > 0.15, and has diminished sensitivity at smaller AOD (Kahn and Gaitley, 2015; Kahn et al., 2010). In this study we use only monthly mean values, for which the uncertainties are expected to be smaller than those for individual retrievals. Note that we did not do a relative humidity (RH) correction to AOD retrievals from MISR as well as other sensors. The seasonal and diurnal variations of AOD represent an integrated effect of variations in aerosol abundance, vertical distribution, chemical constituents, and meteorological conditions.



The MODIS sensors onboard the Terra and Aqua satellites observe the Earth with
multiple wavelength bands over a 2330 km swath (King et al., 2003), which provides near-daily
global coverage. In this study we obtain column AOD data at 550 nm with a $1° \times 1°$ resolution
from the Level 3 monthly atmosphere products Collection 6 (MOD08 and MYD08 for the Terra
and Aqua platforms, respectively). Comparison studies with AERONET have estimated the
accuracy of AOD retrievals to be about $\pm(0.05 + 0.15 \times AOD)$ over land and $\pm(0.03 + 0.05 \times$
$AOD)$ over ocean (Levy et al., 2010; Remer et al., 2005). For both MISR and MODIS data, we
calculate regional mean AOD by averaging valid AOD values over all grids within the three
target regions.
CALIOP is a dual-wavelength polarization lidar on the CALIPSO satellite, and is
designed to acquire vertical profiles of aerosols and clouds at 532 and 1064 nm wavelengths
during both day and night [Winker et al., 2007]. CALIPSO flies in formation with Aqua, and all
three satellites employed in this paper fly in orbits having 16-day repeat cycles. In addition to
vertical extinction profiles, CALIPSO categorizes an aerosol layer as one of seven types based
on a number of parameters including altitude, location, surface type, volume depolarization ratio,
and integrated attenuated backscatter [Omar et al., 2009]. The seven aerosol types are dust,
smoke, clean continental, polluted continental, polluted dust, clean marine, and dusty marine. For
most profiles, this aerosol classification is consistent with that derived from AERONET
inversion data (Mielonen et al., 2009). In this study, we adopt the Level 2 aerosol profile product
(05kmAPro, V4.10), which has an along-track horizontal resolution of 5 km and a vertical
resolution of 60 m or 180 m, depending on whether the aerosol height is below or above 20.2 km
altitude. We do not use the CALIOP Level 3 product because it is difficult to collocate with
AERONET observations (see Section 2.2) due to its coarse resolution ($2° \times 5°$). For each clear-



sky profile, we calculate the column AOD at 532 nm by vertically integrating extinction

coefficients of the features that are identified as "aerosols" and have valid quality control (QC)

flags, i.e., $-100 \leq$ cloud aerosol discrimination (CAD) score $\leq -20$, extinction QC = 0/1, and

extinction coefficient uncertainty < 99.9 (Huang et al., 2013). In addition, we employ two quality

filters used in generating the Level 3 product in order to eliminate features that probably suffer

from surface contamination, i.e., near-surface features with large negative extinction coefficients

and contaminated features beneath the surface-attached opaque layer (NASA CALIPSO team,

2011). Following the same method, we also bin the 532 nm AODs into various height ranges

above ground level (0-200 m, 200-500 m, 500-800m, 800-1200 m, 1200-2000 m, and > 2000 m

above the surface elevation) for the individual aerosol types. Finally, we derive monthly mean

AODs by averaging all clear-sky aerosol profiles within each month over the three target regions.

Although aerosol extinction coefficients within about 200 m of the surface are considered to be

uncertain despite the application of quality filters (NASA CALIPSO team, 2011), we include

them for completeness but exercise with caution when interpreting variations in AODs < 200 m.

It should be noted that CALIPSO AOD is reported at a different wavelength (532 nm) from those

used in the MISR and MODIS products (555 nm and 550 nm, respectively); this slight

wavelength difference is not expected to affect our conclusions regarding AOD seasonal

variations. Another caveat is that the monthly mean AOD from different sensors is calculated

based on different sets of days, since MODIS provides near-daily global coverage while MISR

and CALIPSO do not. We only use in our analysis monthly AOD averaged across 10 years and

all grids/retrievals within three rather large regions (i.e., EUS, WEU, or ECC). In this case, the

impact of the sampling issue is expected to be much smaller than that on the AOD retrieval in an

individual month at a specific location.





2.2 AERONET and surface PM$_{2.5}$ data
We use AOD observations from AERONET to compare with the AOD seasonal
variations derived from satellite datasets. AERONET sunphotometers directly measure AOD at
seven wavelengths (approximately 340, 380, 440, 500, 675, 870, and 1020 nm) with an estimated
uncertainty of 0.01-0.02 (Holben et al., 2001; Eck et al., 1999), which is much smaller than the
uncertainties associated with satellite measurements (Kahn et al., 2010; Levy et al., 2010;
Schuster et al., 2012). Therefore, we consider AERONET as "ground truth" for AOD temporal
variations. We adopt the AERONET Level 2 Version 2.0 direct-sun measurements of spectral
AODs, which are subsequently interpolated to 550 nm using a second-order polynomial fit to
ln(AOD) vs. ln(wavelength) as recommended by Eck et al. (1999). A fundamental difference
between satellite and AERONET AOD observations is that a satellite acquires data at a single
overpass time (or spread over 7 minutes for MISR's nine views) and over an extended spatial
area in the case of MISR and MODIS, whereas AERONET obtains a time series of point data at
each surface station. To match coincident measurements, the AERONET AOD retrievals for
each site are averaged within a 2 h window centered on the satellite overpass times (about 10:30
for MISR and MODIS/Terra, and 13:30 for MODIS/Aqua and CALIPSO, depending on site
location), and compared with the satellite AOD retrievals in a 1° × 1° grid box (consistent with
the grids used in the MODIS Level 3 products) that contains the corresponding AERONET site.
Only those days for which a satellite overpasses an AERONENT site are used in the
comparisons. Since AOD variation has a large spatial correlation length of 40-400 km (Anderson
et al., 2003), spatial averaging over a 1° × 1° grid should not bias the seasonal variations of AOD
but has the benefit of increase the number of data points with valid AOD retrievals that are used
in the comparisons. To assure data quality, only the AERONET sites that span at least 5 years



with at least 10 months of valid data in each year are included in the comparison. After screening,
28, 54, and 13 sites are used in our analysis of the EUS, WEU, and ECC regions.
To provide additional information on the seasonal variations of satellite-observed aerosol
loadings near the surface, we obtain surface $PM_{2.5}$ concentrations from several observational
networks over the three target regions. Hourly $PM_{2.5}$ concentrations for 225 sites over the EUS
region are achieved from the Air Quality System (AQS), which is a large observational database
containing ambient air pollution data collected by the United States Environmental Protection
Agency (USEPA), as well as state, local, and tribal air pollution control agencies in the United
States (USEPA, 2017). For the ECC region, we obtain hourly $PM_{2.5}$ concentrations from the
Ministry of Environmental Protection of China (MEP, http://datacenter.mep.gov.cn/), which
provides continuous measurements at 496 sites located in 74 major cities in China. Hourly/daily
$PM_{2.5}$ concentrations for 52 sites over the WEU region are taken from the European Monitoring
and Evaluation Programme (EMEP). Similar to the processing of AERONET data, we only
include sites whose data span $\geq$ 5 years with $\geq$ 10 months of data in each year, except in the case
of the ECC region where at least 2 years' data are required because the $PM_{2.5}$ concentrations
have been only publicly available since January 2013.
**3 Results and Discussion**
3.1 Seasonal variations of column AOD
Figure 2 illustrates the monthly variations in column AOD observed by MISR,
MODIS/Terra, MODIS/Aqua, and CALIPSO during 2007-2016 in the three target regions. For
consistency with the products from MISR and MODIS, only clear-sky daytime CALIPSO
profiles are used to calculate the lidar-based monthly means. All satellite-borne sensors show
that AOD in the EUS region is the highest in summer and lowest in winter, though CALIPSO





reports a noticeably smaller difference between the summer and winter extrema compared with

the other three satellite instruments. For the WEU and ECC regions, MISR, MODIS/Terra, and

MODIS/Aqua also reveal consistent seasonal patterns in which AOD peaks in spring and/or

summer and reaches its lowest valley in winter. CALIPSO, however, shows little intra-annual

variation in AOD, with small peaks occurring in spring and fall.

In view of the substantial differences between CALIPSO and the other three sensors, we

compare satellite retrieved AOD seasonal variations with point-based ground measurements

from AERONET (Fig. 3). As in other studies, AERONET data are treated as "ground truth" for

column AOD due to its smaller uncertainty compared with satellite data (Kahn et al., 2010; Levy

et al., 2010; Schuster et al., 2012; Fan et al., 2018). Figure 3 shows that, in all three regions, the

AOD seasonal variations measured by AERONET are similar to those retrieved by MISR,

MODIS/Terra, and MODIS/Aqua, but are quite different from CALIPSO data. Reasons for the

differences between CALIPSO and other sensors will be discussed in Section 3.2. Considering

the high accuracy of AERONET, AOD probably peaks in summer/spring and dips in winter. The

higher AOD in summer is probably explained by accelerated formation of secondary aerosols,

including sulfate, nitrate, ammonium, and secondary organic aerosol (SOA), as a result of

stronger radiation and higher temperature in summer. Another possible reason is the higher RH

in summer which favors the hygroscopic growth of aerosols (Liu et al., 2012; Zheng et al., 2017).

While relative pattern of AOD seasonal variations from observations of MISR,

MODIS/Terra, and MODIS/Aqua are similar to each other and to those of AERONET, the

magnitude of AOD observed by these sensors shows remarkable discrepancies. The MISR-

retrieved AOD agrees well with the AERONET observations in EUS and WEU regions (Fig. 3).

In the ECC region, however, MISR underestimates the AERONET AOD, probably because there





is less signal from the surface at higher AOD, which creates ambiguity that can result in the
algorithm assigning too much of the top-of-atmosphere radiance to the surface (i.e., a higher
surface albedo), thereby underestimating the AOD (Kahn et al., 2010). The MODIS/Terra and
MODIS/Aqua significantly overestimate the AERONET AOD in EUS and WEU regions, and
slightly overestimate the AERONET AOD in the ECC region, which is consistent with the
evaluation results of Remer et al. (2005). This overestimation is largely attributed to the
systematic positive bias at low AOD, likely attributed to an instrument calibration issue or an
improper representation of surface reflectance at certain locations and seasons (Remer et al.,
2005). The much smaller overestimation in the ECC region is explained by the fact that the
MODIS AOD is not overestimated or even underestimated at the high AOD range, probably due
to insufficient light absorption in the aerosol models (Remer et al., 2005).
3.2 Seasonal variations of aerosol loadings as a function of height
In addition to column AOD, the climatic effects of aerosols are also strongly dependent
on their vertical distribution. To explore intra-annual variations in aerosol vertical profile, Fig. 4
presents CALIPSO-observed monthly variations of AOD as a function of height in the three
target regions. A striking pattern is that the AOD seasonal variations are dramatically different at
lower and upper heights. Over the WEU and ECC regions, AODs of the vertical layers below
800 m generally peak in winter, while those above 800 m peak in summer/spring. As a result, the
CALIPSO-observed column AOD for these two regions presents a rather uniform seasonal
pattern. For the EUS region, the maximum AOD above 800 m also occurs in summer; however,
AOD below 800 m shows two peaks, one in summer and the other in winter. The integration of
various layers thus yields a nearly unimodal distribution with maximum occurring in summer.



To provide an independent evaluation of the CALIPSO-observed AOD variations at
lower heights, we examine the seasonal variations of near-surface $PM_{2.5}$ concentrations at
hundreds of surface monitor locations within the three target regions. The aerosol extinction
coefficient, and hence AOD at lower heights is affected by not only the particle mass
concentrations, but also aerosol type (absorbing vs. nonabsorbing aerosols, coarse-mode vs. fine-
mode aerosols) and meteorological parameters such as RH, wind speed and direction, and
planetary boundary layer height (Zheng et al., 2017). Nevertheless, previous studies have
reported fairly good correlations between extinction coefficient/low level AOD and $PM_{2.5}$
concentrations (Cheng et al., 2013; Zheng et al., 2017). For this reason, it is reasonable to
qualitatively compare the seasonal variation patterns of near-surface $PM_{2.5}$ concentrations and
low-level AOD. Figure 5 shows that, over the ECC and WEU regions, surface $PM_{2.5}$
concentrations are largest in winter and smallest in summer. In the EUS region, the maximum
$PM_{2.5}$ concentration occurs in summer and a second maximum occurs in winter. These trends are
generally consistent with the seasonal variations of AOD at low heights, implying that CALIPSO
data can generally capture the seasonal changes in low-level aerosol abundance.
The aerosol vertical distribution is an important factor in reconciling CALIPSO and other
sensors with regard to AOD seasonal variations. MISR, MODIS, and AERONET all measure
column-integrated AOD using spectroradiometers, whereas CALIOP is an active lidar which
estimates vertically-resolved AOD based on vertical profiles of attenuated backscatter. By
comparing CALIPSO with the Atmospheric Radiation Measurement (ARM) program's ground-
based Raman lidars, Thorsen et al. (2017) showed that CALIPSO does not detect all relatively
significant aerosols due to insufficient detection sensitivity, and that the fraction of aerosols
detected in the upper air is much smaller than that near the ground surface because the upper-



level aerosols tend to be optically thin. Therefore, the CALIPSO-observed AOD seasonal variations are significantly weighted toward lower heights. Specifically, over WEU and ECC regions, the unimodal AOD distributions with a summer peak at higher levels are largely counteracted by the opposite seasonal variations at lower levels, resulting in rather uniform seasonal variations of column AOD. For the EUS regions, due to the bimodal AOD distribution at lower heights, the summer peak in column AOD variations remain but the difference between peak and valley is smaller than implied by the observations of MISR and MODIS. In this sense, although the integrated CALIPSO column AOD does not agree well with AERONET, it does provide valuable information with respect to intra-annual variations of AOD at specific height ranges.

Why are the AOD seasonal variations different between the lower and upper atmosphere? The atmosphere in winter is generally more stable and vertical mixing is weaker, therefore more aerosols, particularly primary aerosols, are confined to lower heights, resulting in the peak of low-level AOD in winter (Guo et al., 2016; Liu et al., 2012; Zheng et al., 2017). At higher levels, the maximum AOD in summer can be explained by two reasons: (1) more aerosols, especially primary aerosols, are transported to the upper level in summer due to stronger vertical mixing (Guo et al., 2016; Liu et al., 2012; Zheng et al., 2017), and (2) secondary aerosol formation is more rapid in summer because of stronger radiation and higher temperature, and much of the secondary aerosols are produced in the upper air (de Reus et al., 2000; Minguillon et al., 2015; Heald et al., 2005). In addition, the seasonal variations of AOD at different vertical levels may also be influenced by the variations of RH which affects the hygroscopic growth (Liu et al., 2012; Zheng et al., 2017) as well as the seasonal patterns of inter-regional transport of aerosols (Tian et al., 2017).



3.3 Seasonal variations of aerosol types
Besides column AOD and vertical profiles, another factor influencing aerosol climate
impact is aerosol type (i.e., partitioning by size and chemical composition). The MISR and
CALIPSO products classify aerosols based on distinct principles of measurement and retrieval
algorithms. Analysis of the two datasets in combination can potentially lead to a deeper
understanding of the factors driving temporal variations of aerosol type. Figures 6 and 7 illustrate
the seasonal variations of AODs for various aerosol types retrieved by MISR and CALIPSO,
respectively. As discussed in Section 3.2, relative variability in CALIPSO-derived AOD at
different height ranges appears to be more reliable than integrated column AOD, therefore we
show aerosol types below and above 800 m separately in Fig. 7.
MISR distributes AODs into three size ranges, i.e., small (< 0.7 μm diameter), medium
(0.7-1.4 μm diameter), and large (> 1.4 μm diameter). Among the major constituents of ambient
aerosols, which include primary aerosols (dust, sea-spray aerosols, and primary anthropogenic
aerosols) and secondary aerosols (sulfate, nitrate, ammonium, and SOA), dust and sea-spray
aerosols are predominantly coarse particles and secondary aerosols are dominated by very fine
particles, while primary anthropogenic aerosols span a large size range, leading to a mean size
intermediate between dust/sea-spray and secondary constituents (Seinfeld and Pandis, 2006). Fig.
6 indicates that the small-size AOD is much larger in spring/summer than in winter over all
regions, whereas large-size AOD generally shows rather uniform distributions, except for the
ECC region where a peak occurs in late winter/early spring. The high small-size AOD in summer
is probably due to accelerated secondary aerosol formation and enhanced hygroscopic growth, as
described in Section 3.1. In contrast, AOD of primary anthropogenic aerosols should be less
influenced by seasonal effects, which partly accounts for the rather uniform distributions of





large-size AOD. Additionally, the changes in large-size AOD are also affected by other aerosol

components including dust and sea-spray aerosols, as discussed below.

In contrast to MISR's partitioning of aerosol type by size, shape, and absorption, the

CALIPSO-retrieved aerosol types (Fig. 7) are characterized by emission source. Particles

associated with anthropogenic air pollution (polluted continental and polluted dust) comprise the

dominant type in all three regions. At higher levels, the maximum AOD of polluted

continental/dust aerosols occurs in spring/summer, while the maximum occurs in winter at lower

levels (plus a second maximum in summer in EUS), in accordance with the seasonal variations

of total AOD at different heights, as discussed in Section 3.2. With regard to dust and clean

marine (sea-spray) aerosols, the AOD in the EUS region does not show an obvious seasonal

pattern. In the WEU region, AOD of dust aerosols peaks in summer, consistent with previous

surface-based observational studies which show that dust events in Europe predominantly occur

during summer due to transport from the Sahara region (Stafoggia et al., 2016). The AOD of dust

above 800 m is much larger than that below 800 m, supporting the conclusion that dust aerosols

in WEU mainly originate from long range transport. Since the dust AOD has a quite large inter-

annual variation (denoted by the error bars in Fig. 7), we use the Student's t-test to demonstrate

the statistical significance of the seasonal variations shown above. The dust AOD in summer is

statistically larger than that in any other season at the 0.05 level, indicating the robustness of the

peak in summer. Contrary to dust, the AOD of sea-spray aerosols in WEU is much higher in

winter than in summer, probably because winter is the relative windy season with large low

pressure systems over the Atlantic Ocean and the North Sea (Manders et al., 2009). The offset of

the opposite variation trends in dust and sea-spray aerosols partly accounts for the rather uniform

distributions of large-size AOD (see Fig. 6). Over the ECC region, sea-spray aerosols make a





negligible contribution to total AOD. The dust AOD is much larger in spring than in any other

season (significant at the 0.05 level), which is tied to the outburst of springtime Gobi desert dust

storms (China Meteorological Administration, 2012). The high dust AOD explains the peak in

large-size AOD in spring over the ECC region (see Fig. 6).

Smoke aerosols are predominantly located above 800 m. Over the EUS and WEU regions,

smoke aerosols present a unimodal distribution with maximum occurring in summer. The

differences between smoke AOD in summer and the other three seasons are all statistically

significant at the 0.05 level, except for the difference between summer and spring over the WEU

region, which is statistically significant at the 0.10 level. In the ECC region, the smoke AOD

follows a bimodal distribution with peaks occurring in March and August and valleys occurring

in May and December. The differences between either of the peak months and either of the

valley months are statistically significant at the 0.05 level. MISR's independent retrieval of

absorbing AOD (Fig. 6) presents a similar seasonal pattern (statistically significant at the 0.05

level) as the CALIPSO smoke AOD. In fact, smoke consists of a larger fraction of absorbing

aerosols (Dubovik et al., 2002), such as BC and light-absorbing organic aerosol (Kirchstetter and

Thatcher, 2012), as compared to other aerosol types. The variability of MISR absorbing AOD

(shown in the right Y-axis of Fig. 6) is about 0.002-0.005, while the variability of smoke AOD

from CALIPSO is about 0.01-0.03. The smoke AOD includes the contributions of both the

absorbing and scattering portions. The MISR absorbing AOD, which is calculated using total

AOD × (1 − single scattering albedo), represents only the absorbing portion but includes

contributions from aerosol types other than smoke (Bull et al., 2011). Considering that the single

scattering albedo of smoke is about 0.80-0.94 (Dubovik et al., 2002), the MISR absorbing AOD

and CALIPSO smoke AOD are consistent in the order of magnitude. For the preceding reasons,



the seasonal patterns of smoke and absorbing aerosols acts as a cross-validation and strengthens
the reliability of the observed trends. Over the EUS and WEU regions, the largest smoke AOD in
summer could be explained by the highest emissions from forest and grassland fires (van der
Werf et al., 2017). Over the ECC region, an additional peak occurs in March because agricultural
residue burning makes a substantial contribution to total smoke emissions (van der Werf et al.,
2017), and such burning takes place more frequently in March due to burning of crop residues
left on the fields from the previous growing season (Shon, 2015).

3.4 Diurnal variations of height- and type-resolved AODs

CALIPSO provides both daytime and nighttime aerosol retrievals, giving an opportunity

to analyze aerosol diurnal variations. Care must be exercised in interpreting the CALIPSO-
observed diurnal variability because CALIPSO's detection sensitivity is lower during daytime
due to interference from sunlight. As a result, the difference between daytime and nighttime
AODs represents an overall effect of both actual and artificial diurnal variability. In this section,
we identify variation patterns that are most likely real, in spite of the possible effects of
measurement artifacts.

Figure 8 shows the differences between nighttime and daytime total and height-resolved

AODs (nighttime minus daytime values). The sign of the difference depends on the relative
importance of two competing factors: a more stable atmosphere at night favors the accumulation
of aerosols, whereas stronger photochemical reactions enhance aerosol loading during the day.
Fig. 8 reveals that nighttime AODs tend to be smaller than daytime AODs in winter. In contrast,
during summer and/or spring, nighttime AODs are higher than (WEU and ECC) or similar to
(EUS) the daytime values. More precisely, the difference between nighttime and daytime AODs
is significantly more positive in summer. This pattern is applicable to both AODs < 800 m and >



800 m. It is unlikely to be attributable to the difference in CALIPSO's detection sensitivity,
which presumably exerts similar effects in all seasons. It should be noted, however, that we
cannot fully exclude the effects of measurement artifacts, the quantification of which require
further investigations using more accurate measurement techniques. A probable reason for more
positive nighttime-daytime AOD difference in summer is that the diurnal temperature range,
which is defined as the difference between the maximum and minimum temperatures in a day, is
larger in summer (Ruschy et al., 1991; Jackson and Forster, 2010; Sun et al., 2006), giving rise to
a larger difference in nighttime and daytime AODs due to a more stable nighttime atmosphere
compared to a more unstable daytime atmosphere.

We further illustrate the differences between nighttime and daytime AODs of major

aerosol types (Fig. 9). Smoke AOD is much higher in the nighttime than in the daytime.
Considering that the emission rates of smoke aerosols is likely to be similar at different time of
the day, the higher nighttime smoke AOD is probably the result of increased atmospheric
stability at night, allowing the aerosols to accumulate. On the contrary, dust AOD is usually
higher in the daytime, which may be tied to higher near-surface wind speed in the day (He et al.,
2013; He et al., 2012; Hasson et al., 1990). The diurnal variations of polluted continental/dust
aerosols vary according to season and height, and likely depend on the relative roles of more
stable atmosphere at night and more active chemical reactions in the day. The type-dependent
diurnal variations should mainly be representative of actual conditions, as such different
variations as a function of aerosols type are unlikely explained by instrument detection
sensitivity.



**4 Conclusions and implications**


This study investigated the seasonal and diurnal variations of aerosol column loading,
vertical distribution, and particle types using multiple satellite and ground-based observational
datasets during 2007-2016 over EUS, WEU, and ECC regions. Retrievals from MISR and
MODIS reveal that column AOD in all three regions peaks in spring/summer and reaches its low
in winter, which is consistent with observations from AERONET. This seasonal pattern is
probably explained by accelerated formation of secondary aerosols in spring/summer due to
stronger insolation and higher temperature. In contrast, CALIPSO shows a much weaker
seasonal variability in column AOD, probably because CALIPSO-retrieved AOD is weighted
toward lower heights since some thin aerosol layers in high levels are undetected due to
insufficient detection sensitivity. Despite the discrepancy in integrated column AOD, CALIPSO
does provide valuable information with respect to intra-annual variations of AOD as a function
of height. Over the WEU and ECC regions, AODs of the vertical layers below 800 m generally
peak in winter, while those above 800 m mostly peak in summer. For the EUS region, the
maximum AOD above 800 m also occurs in summer; however, AOD below 800 m shows two
peaks, one in summer and the other in winter. The seasonal variations of AOD at low heights are
consistent with seasonal patterns of measured surface $PM_{2.5}$ concentrations.
When aerosols are binned into different size ranges, the small-size AOD is much larger in
spring/summer than in winter over all three regions. Large-size AOD generally shows rather
uniform distributions, except for the ECC region where a peak occurs in spring, consistent with
the largest dust AOD in this season. When aerosols are classified according to sources, the
aerosols associated with anthropogenic air pollution (as well as mixtures of anthropogenic
pollution and dust) are the dominant type in all three regions. AOD of polluted aerosols has a

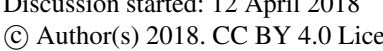


similar seasonal pattern as total AOD. Dust and clean marine aerosols in the WEU region peak in

summer and winter, respectively, whereas they do not show an obvious seasonal pattern in the

EUS region. Smoke aerosols, which CALIPSO indicates are predominantly located at heights

above 800 m, present an obvious unimodal distribution with maximum occurring in summer over

EUS and WEU regions, and a bimodal distribution with peaks in August and March over the

ECC region. This pattern is in good agreement with the seasonal variations of absorbing AOD

derived from MISR.

Regarding diurnal variations, the difference between nighttime and daytime AODs

(nighttime minus daytime) is more positive in summer than in winter, which is likely explained

by larger diurnal temperature range in summer. Smoke AOD is much higher in the nighttime,

when the atmosphere is more stable, than in the daytime. On the contrary, dust AOD is usually

higher in the daytime, when higher winds speeds enable a greater abundance of particles to

become airborne.

The combination of multiple satellite and ground-based observations facilitate a

systematic and deeper understanding of the seasonal and diurnal variations of aerosols,

particularly their vertical and type distribution. Comparison of multiple measurement and

retrieval methodologies enables reducing the uncertainties in the estimation of aerosol direct

effects by providing improved information about aerosol vertical and type distributions, which

significantly affect the aerosol-induced scattering and absorption of radiation. More importantly,

the intra-annual variations of vertical distributions and types of aerosols are important for

understanding their impact on atmospheric dynamics, cloud fields, and precipitation production.

Many studies (Chen et al., 2017; Bond et al., 2013) have shown that BC and dust can either

enhance or inhibit convection and hence cloud fields, depending on their vertical locations,





which are very different from the effects of non-absorbing aerosols (Ramanathan et al., 2005;
Fan et al., 2008; Massie et al., 2016). Finally, the data and variation patterns presented in this
study can be used to evaluate and improve model simulations, with the ultimate goal of
improving model assessment of the climatic and health effects of aerosols.

## Acknowledgments

This study was supported by the MISR project at the Jet Propulsion Laboratory,
California Institute of Technology, under contract with NASA, NASA CCST program, and NSF
AGS-1701526. We acknowledge Michael J. Garay, Jason L. Tackett, and Ali H. Omar for their
valuable comments and suggestions. All data needed to evaluate the conclusions are present in
the paper.

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





**Figures**

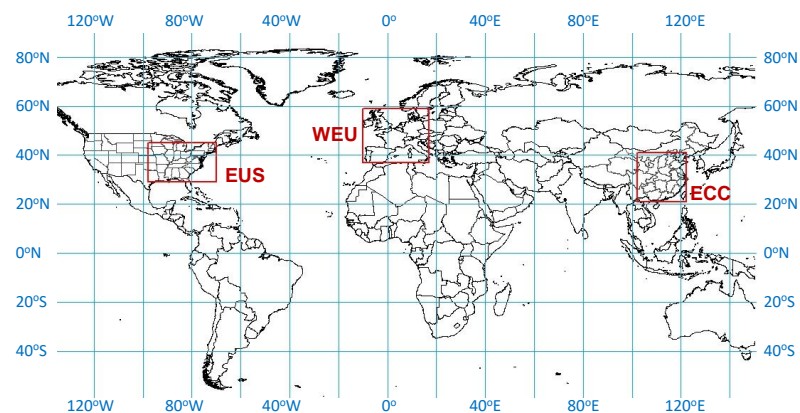

**Figure 1**. Target regions for this study: the Eastern United States (EUS), Western Europe
(WEU), and Eastern and Central China (ECC).

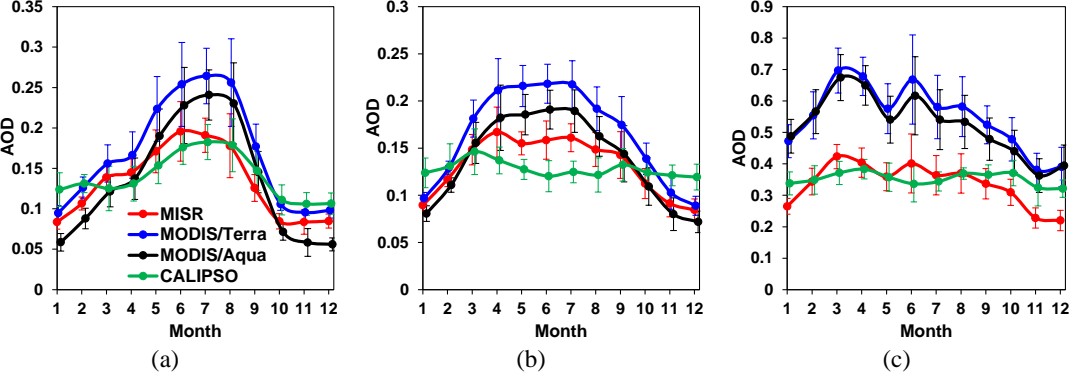

|  |  |  |
|---|---|---|
| (a) | (b) | (c) |

**Figure 2**. Monthly mean AOD observed by MISR, MODIS/Terra, MODIS/Aqua, and CALIPSO
during 2007-2016 in (a) EUS, (b) WEU, and (c) ECC. For CALIPSO, only clear-sky daytime
profiles are averaged in order to be consistent with the MISR and MODIS products. The error
bars denote the standard deviation of the monthly mean AOD values obtained over all years.
Note the different scales on the y-axes of the plots.



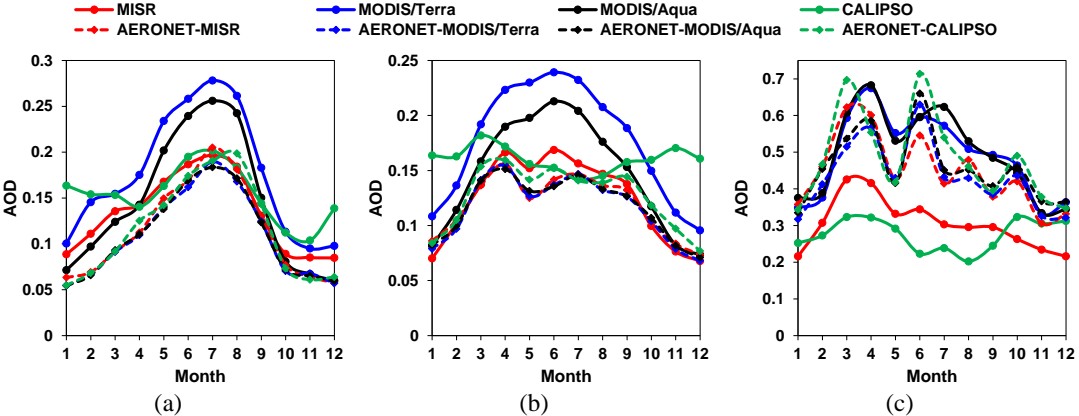

(a)  (b)  (c)

**Figure 3.** Monthly mean AOD observed by satellites and AERONET averaged across the AERONET sites during 2007-2016 in (a) EUS, (b) WEU, and (c) ECC. The observations from MISR, MODIS/Terra, MODIS/Aqua, and CALIPSO are averaged over 1°×1° grid boxes containing the AERONET sites. The AERONET data are averaged within a 2 h window centered on satellite overpass times. The numbers of AERONET sites included in analysis are 28, 54, and 13, in the EUS, WEU, and ECC regions, respectively. Since the four sensors overpass a site in different days and different times of day, we separately calculate the AERONET data matched to each sensor (denoted by "AERONET-×××"). The AERONET curves matched to different sensors are close in EUS and WEU, partly because there are plenty of sites in these two regions, and the discrepancy due to the sampling issue is therefore smoothed out. In contrast, there are only 13 AERONET sites in ECC, so there exists larger discrepancy between the AERONET data matched to different sensors. Note the different scales on the y-axes of the plots.





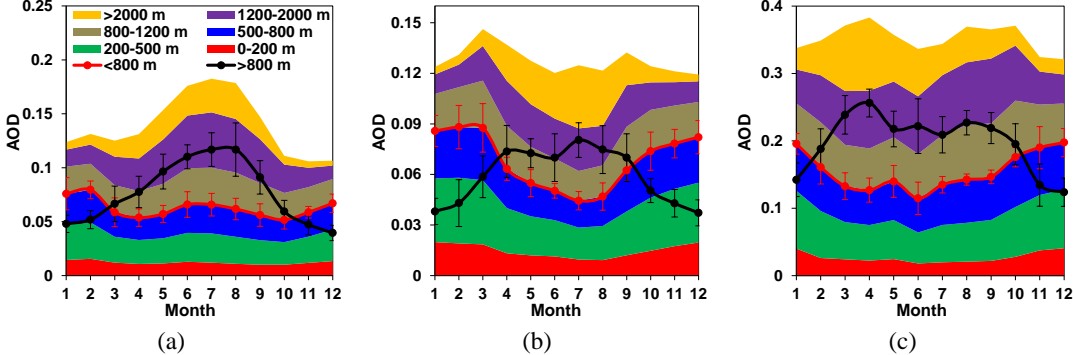

<p style="text-align:center">(a)   (b)   (c)</p>

**Figure 4**. Monthly mean AOD as a function of height above ground level observed by CALIPSO during 2007-2016 in (a) EUS, (b) WEU, and (c) ECC. Only clear-sky daytime profiles are averaged in order to be consistent with the products of MISR and MODIS. The range of AOD within a particular height range is depicted by the colored stacks. The integrated AODs for heights below and above 800 m are shown as solid lines, for which the error bars are defined in the same way as in Fig. 2. Note the different scales on the y-axes of the plots.

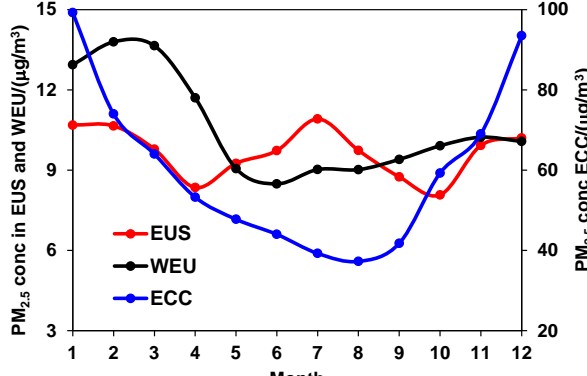

**Figure 5.** Monthly mean surface $PM_{2.5}$ concentrations during 2007-2016 in three target regions. The numbers of observational sites included in averaging are 225, 52, and 496, in the EUS, WEU, and ECC regions. Note the different scales on the y-axes for EUS/WEU and ECC.



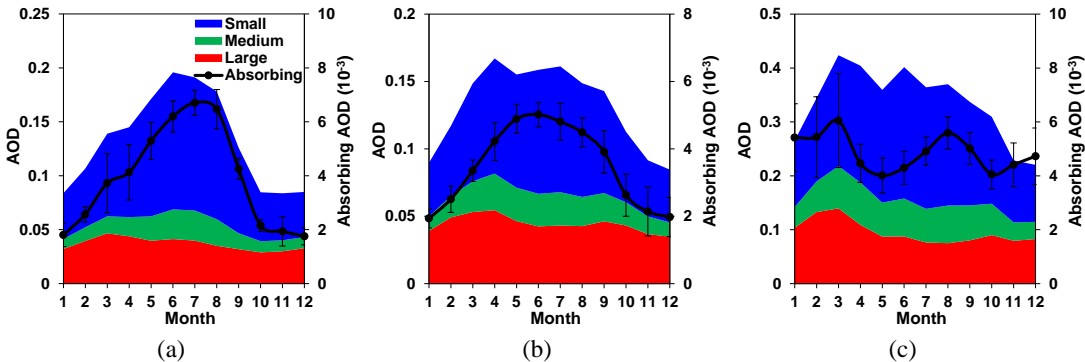

**Figure 6**. Monthly mean AOD of different aerosol types observed by MISR during 2007-2016 in
(a) EUS, (b) WEU, and (c) ECC. The size-resolved AODs are depicted by the colored stacks
(left Y-axis); the integration of the three size ranges yields total column AOD, as represented by
the upper edge of the blue color. The AOD of absorbing aerosols is shown as solid lines (right
Y-axis), for which the error bars are defined in the same way as in Fig. 2. Note the different
scales on the y-axes of the plots.





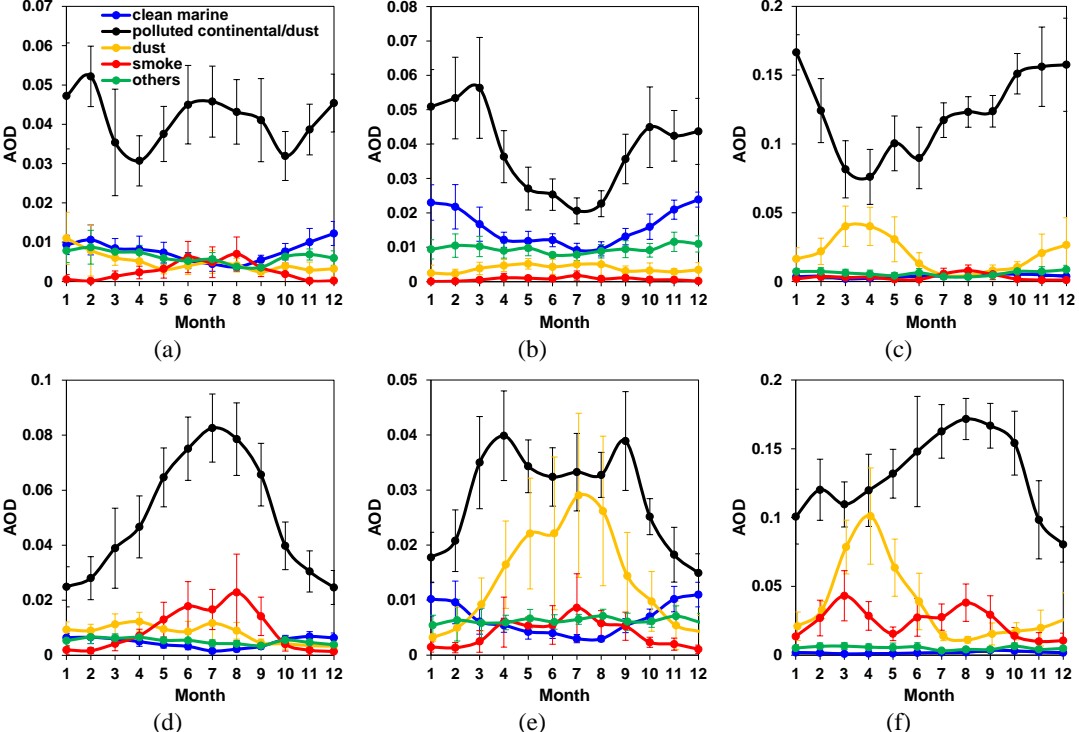

**Figure 7**. Monthly mean AOD of different aerosol types (a-c) below 800 m and (d-f) above

800 m observed by CALIPSO during 2007-2016 in (a, d) EUS, (b, e) WEU, and (c, f) ECC. Only

clear-sky daytime profiles are used in the averaging to be consistent with the products of MISR

and MODIS. The definition of error bars is the same as in Fig. 2. Note the different scales on the

y-axes of the plots.





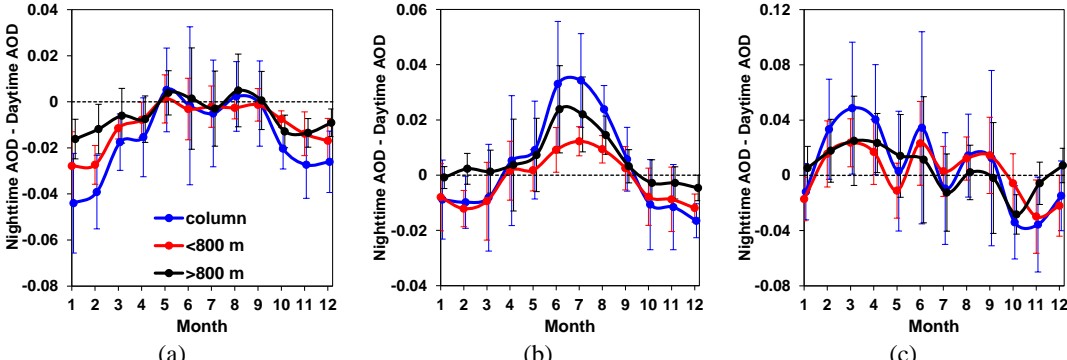

**Figure 8**. Differences between monthly mean nighttime and daytime AODs (including column AODs, as well as AODs of aerosols located below 800 m and above 800 m) observed by CALIPSO during 2007-2016 in (a) EUS, (b) WEU, and (c) ECC. Only clear-sky profiles are included. The definition of error bars is the same as in Fig. 2. Note the different scales on the y-axes of the plots.



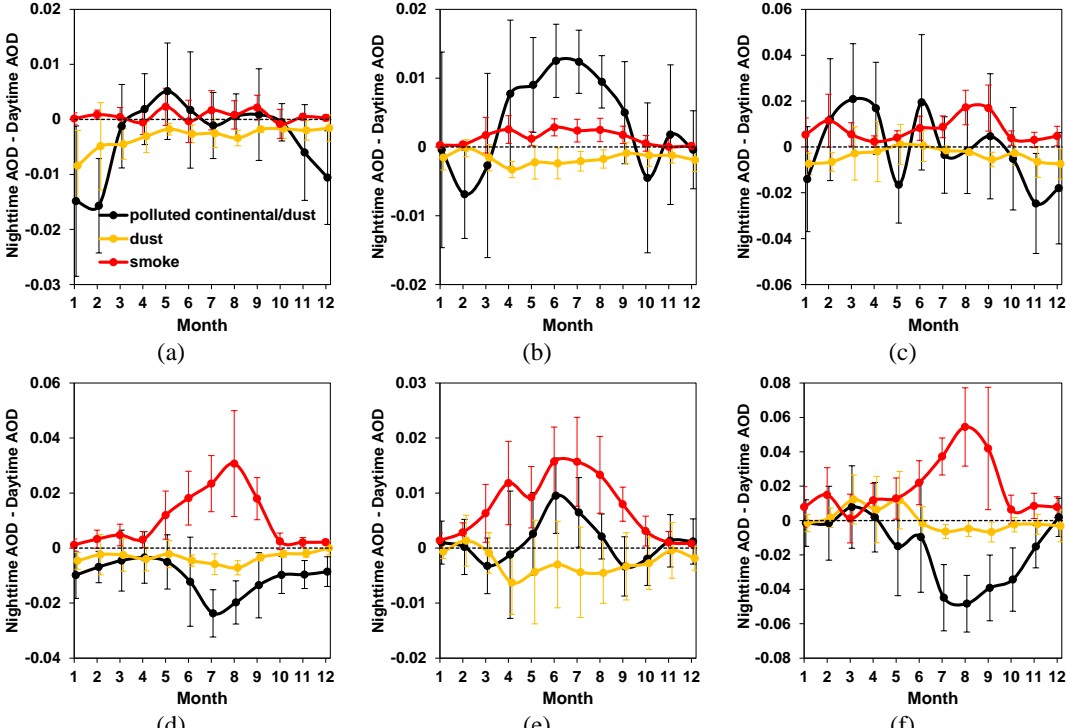

**Figure 9**. Differences between monthly mean nighttime and daytime AODs of different aerosol

types (a-c) below 800 m and (d-f) above 800 m observed by CALIPSO during 2007-2016 in (a,

d) EUS, (b, e) WEU, and (c, f) ECC. Only clear-sky profiles are included. The definition of error

bars is the same as in Fig. 2. Note the different scales on the y-axes of the plots.