# Peer review of "particle types from multiple satellite and ground-based observational datasets"

_Atmospheric Chemistry and Physics, 2018_

## Referee Comment (RC1) · Anonymous Referee #1 · 8 May 2018

Aerosol radiative effect is a hot topic in the science community, which is dependent on the aerosol optical properties, size distribution, aerosol types, and their vertical distribution. While many studies have examined the seasonal and diurnal variations of aerosols, few studies have examined the vertical distributions of aerosol amount and types, which can strongly affect the aerosol radiative effect and corresponding thermal impacts on the profiles of temperature. Using multi-satellite observations along with the surface observations of aerosols, this study examines the the seasonal and diurnal variations of aerosol column loading, vertical distribution, and particle types over

three populous regions: the Eastern United States (EUS), Western Europe (WEU), and Eastern and Central China (ECC). Interesting statistical characteristics about the dominant aerosol types and vertical distributions have been obtained. The paper is also organized and written well.

Detailed Comments:

Line 45-48: I am not sure if the climate effects of $CO_2$ also depends on its intra-annual variability, particularly considering the higher outgoing longwave radiation from surface in summer than in winter.

Line 49-54: Not only the scattering and absorption properties, but also the size distribution and vertical distribution of aerosols can also cause problems. Zheng et al. (2017, ACP) have shown that the vertical distribution of AOD could have strong impacts on the aerosol concentration (mass concentration) within PBL, which directly affect cloud properties and then change cloud radiative forcing. Garrett et al. (2004, GRL) showed that nucleation mode aerosols and accumulation mode aerosols have very different scattering radiative effects.

Line 54-58: Besides to the TOA radiative balance, convective clouds development, absorptive and non-absorptive aerosols have different impacts on the surface radiative cooling effects, as shown by Yang et al. (2016, JGR) which demonstrated that more absorptive aerosol can cause more surface cooling effects.

Line 59-63: The vertical distributions are also important for aerosol-cloud interaction since only the aerosols near cloud bases have strong interaction with cloud properties. Zhao et al. (2018, Earth and Space Science) showed Twomey effect using in-situ aircraft observations in East China instead of anti-Twomey effect found using column aerosols based on satellite observations.

Line 101-103: You may indicate the data observation time, such as day time and night time.

Line 158: I would suggest "variations of AODs with heights below 200 m"

Line 162-167: The different time representation errors could be noticeable for monthly average. For example, Wang and Zhao (2017, JGR) showed that the MODIS cloud time representation errors could be up to 3-4% for monthly average (10 years data) while much smaller for yearly average and much larger for daily average. However, this might not affect the findings/conclusions of this study. You may simply indicate/inform the noticeable time representation error of 3-4% (Wang and Zhao, 2017) while what they studied are cloud coverage instead of aerosols.

Line 267-269: I highly agree with this analysis. By doing this, the effects of PBL and relative humidity could be minimized.

Line 303-304: Yang et al. (2018, AR) also showed the winter vs summer patterns (high in winter, low in summer) of inter-regional transport (between pearl river delta and Hongkong) of aerosols; Garrett et al. (2010, Tellus B) also showed the strong transport in winter and weak transport in fall for aerosols from mid-latitudes to Arctic. Actually, the vertical distribution of aerosols is very interesting in the Arctic, with high values at heights around 2-7 km, mainly caused by long-range transport from other regions such as mid-latitudes.

Line 374, acts -> act, strengthens -> strengthen

Line 400-405, Since AOD includes the impacts of relative humidity, is it possible that relative humidity also contributes to the diurnal variation of AOD? Another possible contribution might be the growth of fine aerosols, as indicated by Zhao et al. (2018, AAS), the growth rates of fine mode aerosols generally starts from the morning time with a growth rate of around 2-7 nm/hour (1.7-6.5 nm/hour near Beijing region in summer). They could become accumulation mode from fine mode aerosols at night time or on next day (then day time) making AOD larger.

Line 462-465: I agree that absorbing aerosols could play different roles to convection

and clouds from non-absorbing aerosols. However, I think both absorbing aerosols and non-absorbing aerosols will reduce the direct solar radiation reaching the surface, causing surface cooler and further affect convection and clouds in similar way (of course, no absorption of solar radiation in the air), while the effects could be weaker than absorbing aerosols.
* * *

---

## Referee Comment (RC2) · Anonymous Referee #2 · 11 May 2018

Title: Intra-annual variations of regional aerosol optical depth, vertical distribution, and particle types from multiple satellite and ground-based observational datasets

Summary: The paper combines retrievals and observations from multiple satellites (active and passive) and ground based (in-situ and remote sensed), in order to characterize the seasonal and diurnal variations of aerosol properties in three heavily-populated regions (EUS, WEU and ECC). The aerosols are separated into lower (< 800 m) and higher levels (> 800m), monthly averages are calculated, and annual cycles plotted. Analysis and interpretation and some speculation are presented.  The main conclusions are that in all three regions, column AOD and higher level AOD all peak in the summer, whereas AOD in lower levels peaks during the winter. AOD from fine-sized particles peaks in the spring/summer and is attributed to anthropogenic sources. Dust and sea-salt peaks in the winter in WEU but are nearly constant in the other two regions. There appears to be larger nighttime/daytime AOD difference in summer than winter.

This paper is logical and easy to read. The English language usage is satisfactory. The idea of separating into low level (e.g. < 800 m; presumably a proxy for boundary layer) and higher level (> 800 m) is unique. I wish I could believe all of the conclusions. But I don't yet. Like I explained in the "initial quality" review, I have strong concerns about data sampling. For example, Colarco et al., (2014, [10.5194/amt-7-2313-2014]) explains that "sampling matters", and that when we develop climatology from different types of orbits (and coverage), we need to deal with this problem. Because of this, I don't think that "the impact of the sampling issue is expected to be much smaller than that on the AOD retrieval in an individual month at a specific location" (Lines 162-167). If MODIS calculates monthly mean based on all 30 days and CALIPSO based on 2-4 times month (every 8 or 16 days, if lucky), then we don't expect the monthly means to match. Of course, if there are clouds, this could be MODIS making monthly means from, say 10 days, and CALIPSO making only one. One more paper to think about is Chin et al., (2014; [10.5194/acp-14-3657-2014]). Although they study multi-year data, they make points about comparing datasets with all kinds of sampling differences.

Of course, the low bias (Fig 2, lines 216-218) between CALIPSO -derived AOD and the other satellites (MODIS, MISR), can be because of assumed lidar ratios (Ma etal., AMT; [10.5194/amt-6-2391-2013]), or undetected aerosol layers (Kim et al., JGR, [10.1002/2016JD025797]). I guess that the Thorson (2017) reference already listed could be a reason as well.

Why the non-confident statements ("probably") in lines 227-231? I think you should be able to find references. What are the sources of these SOA? What about long-range

transcript in the summer?

Considering lines 235-236, I again ask about sampling? Are the monthly means for AERONET and satellites computed based on the same days? Or is mean AERONET = mean (of AERONET data) and mean satellite = mean (of SATELLITE data)? We know from validation exercises that when actually collocated in space and time (both AERONET and satellite are free of clouds) that they match overall well (yes, sometimes small biases, e.g. Remer et al., 2005). However, I do not expect matches if using different samples (days). Note that the Remer et al., (2005) study has been updated for MODIS (Collection 5 and Collection 6), and there are also updates for MISR evaluations. The "instrument calibration issue" (lines 243-244) would not cause such a large bias.

I think it is a good idea that you are comparing low-level CALIPSO to ground level PM2.5 (lines 267-269) but I wonder about the temporal sampling. Also, PM2.5 is usually a "dried" aerosol measurement whereas CALIPSO is ambient RH.

I don't understand the arguments in lines 279-282, in that since CALIPSO can't detect thin aerosols, that the fraction of upper-level aerosols is smaller than at the surface, and that results in the CALIPSO AOD as being weighted toward lower heights. According to the Kim et al., paper (listed above), CALIPSO is likely to miss stuff close to the ground. Anyway, the point is I don't think you can say that CALIPSO is missing stuff, and yet it "provides valuable information with respect to intra-annual variations at specific height ranges" (line 289-290).

Thank you for adding many references in lines 292-304 to discuss why AOD seasonal differences should be different in lower versus higher altitudes. I don't know if I agree that "seasonal variations of AOD at different levels are influenced by variations of RH which affects hygroscopic growth" (Lines 301-303). Of course, RH influences AOD, but it is total column water vapor and not necessarily RH that changes drastically from season to season.

I agree that comparing MISR-derived aerosol "types (size/ absorption)" and CALIPSO-derived aerosol "types" (sources) is ultimately useful. (lines 307-308). However, they are clearly different beasts, and I am getting lost reading this section (Section 3.3). Each paragraph has multiple sentences that are "A implies B, whereas (while / in contrast) sometimes C implies D". It's hard to follow. I suggest a table, or schematic cartoons, or bullets.

I notice that regarding the aerosol typing as seen by CALIPSO, all of the regions (over land), have non-trivial amount of "clean-marine" aerosol (Fig 7). Is this transported marine aerosol to the entirety of the regional box, or should the marine aerosol be expected to be more dominant but confined only to the coastal areas of a region?

The section on daytime/nighttime variability is nice, but I think it is beside the point of the rest of the paper. Why would smoke AOD accumulate at night? Higher RH at night might make bigger aerosols, but if anything, fire activity is reduced at night. You might check the PM2.5 measurements here. I suggest leaving this section out, and thinking about the questions related to the other sections. "Intra-annual", "vertical", and "particle types" is enough for one paper!

In terms of figures. I can see why the authors do this (different magnitudes of AOD and or PM2.5 at different sites), but the varying y-axes within figure captions, and from figure-to-figure are distracting. But thank you for pointing out in the caption!

What is "upper air"? I see it a few places, and assume you mean > 800 m AGL? (e.g. Line 300).

The abstract suggests (lines 37-38) that results can "help to improve the current estimates of climatic and health impacts of aerosols". Well maybe, but I would drop this from the abstract since there is no discussion in the paper.

---

## Author Comment (AC1) · 12 Jun 2018

We thank the reviewer for the valuable comments. We have followed these comments in revising the manuscript. Please see attached for a point-to-point response and the revised manuscript.

Please also note the supplement to this comment: https://www.atmos-chem-phys-discuss.net/acp-2018-110/acp-2018-110-AC1-supplement.zip

---

## Author Response (AR1)

Reviewer 1:

Aerosol radiative effect is a hot topic in the science community, which is dependent on the aerosol optical properties, size distribution, aerosol types, and their vertical distribution. While many studies have examined the seasonal and diurnal variations of aerosols, few studies have examined the vertical distributions of aerosol amount and types, which can strongly affect the aerosol radiative effect and corresponding thermal impacts on the profiles of temperature. Using multi-satellite observations along with the surface observations of aerosols, this study examines the seasonal and diurnal variations of aerosol column loading, vertical distribution, and particle types over three populous regions: the Eastern United States (EUS), Western Europe (WEU), and Eastern and Central China (ECC). Interesting statistical characteristics about the dominant aerosol types and vertical distributions have been obtained. The paper is also organized and written well.

Response: We thank the reviewer for the valuable comments. We have followed these comments in revising the manuscript. Point-to-point responses are given below.

Detailed Comments:

Line 45-48: I am not sure if the climate effects of CO2 also depends on its intra-annual variability, particularly considering the higher outgoing longwave radiation from surface in summer than in winter.

Response: We no longer mention the climate effects of $CO_2$ in the revised manuscript. The revised sentence is as follows:

Therefore, the climatic and health effects of aerosols are not only induced by inter-annual concentration changes, but also strongly depend on their intra-annual variability. (Line 41-43)

Line 49-54: Not only the scattering and absorption properties, but also the size distribution and vertical distribution of aerosols can also cause problems. Zheng et al. (2017, ACP) have shown that the vertical distribution of AOD could have strong impacts on the aerosol concentration (mass concentration) within PBL, which directly affect cloud properties and then change cloud radiative forcing. Garrett et al. (2004, GRL) showed that nucleation mode aerosols and accumulation mode aerosols have very different scattering radiative effects.

Response: Following this comment, we have described the impact of size distribution and vertical distribution and cited these two references. The revised text is as follows:

However, the wide ranges of particle optical properties and size distribution mean that even for the same AOD, different aerosol components have different effects on not only the magnitude, but also the sign, of aerosol radiative forcing (IPCC, 2013; Gu et al., 2006; Garrett et al., 2004). (Line 46-49)

Besides aerosol type, the aerosol vertical distribution influences its mass concentration within the planetary boundary layer (PBL) (Zheng et al., 2017) and the vertical profile of heating rate (Johnson et al., 2008; Guan et al., 2010; Zhang et al., 2013), which subsequently modifies the atmospheric stability and convective strength (Ramanathan et al., 2007), with potential changes in cloud properties (Johnson et al., 2004). (Line 54-58)

Line 54-58: Besides to the TOA radiative balance, convective clouds development, absorptive and non-absorptive aerosols have different impacts on the surface radiative cooling effects, as shown by Yang et al. (2016, JGR) which demonstrated that more absorptive aerosol can cause more surface cooling effects.

Response: We have mentioned the different impacts of absorptive and non-absorptive aerosols on surface radiative cooling effects in the revised manuscript:

Furthermore, absorbing and non-absorbing aerosols have been found to have very different impacts on the surface radiative cooling effects (Yang et al., 2016) and the development of convective clouds (Massie et al., 2016; Ramanathan et al., 2005; Rosenfeld et al., 2008). (Line 51-54)

Line 59-63: The vertical distributions are also important for aerosol-cloud interaction since only the aerosols near cloud bases have strong interaction with cloud properties. Zhao et al. (2018, Earth and Space Science) showed Twomey effect using in-situ aircraft observations in East China instead of anti-Twomey effect found using column aerosols based on satellite observations.

Response: Following the reviewer's comments, we have added this point in the revised manuscript:

Understanding aerosol variability as a function of height is also important because the indirect effect of aerosols is mainly dependent on those mixed with the clouds (Zhao et al., 2018b). (Line 59-60)

Line 101-103: You may indicate the data observation time, such as day time and night time.

Response: We have added the following description in the revised manuscript:

The aerosol retrievals from MISR and MODIS are only available for clear-sky conditions in the daytime. CALIPSO provides retrievals during both day and night, but only clear-sky daytime profiles are used in order to be consistent with the products from MISR and MODIS. (Line 99-102)

Line 158: I would suggest "variations of AODs with heights below 200 m"

Response: Done. Thank you! (Line 155)

Line 162-167: The different time representation errors could be noticeable for monthly average. For example, Wang and Zhao (2017, JGR) showed that the MODIS cloud time representation errors could be up to 3-4% for monthly average (10 years data) while much smaller for yearly average and much larger for daily average. However, this might not affect the findings/conclusions of this study. You may simply indicate/inform the noticeable time representation error of 3-4% (Wang and Zhao, 2017) while what they studied are cloud coverage instead of aerosols.

Response: In the revised manuscript, we have investigated the impact of spatiotemporal sampling on seasonal variations of AOD using two sensitivity scenarios, and cited this reference. The added text is shown as follows:

As described in Section 2.1, MODIS provides near-daily global coverage but MISR and CALIPSO do not. As a result, the monthly mean AOD from different sensors is calculated based on different sets of days, which might lead to uncertainties in the estimation of monthly mean AOD (Colarco et al., 2014; Wang and Zhao, 2017). To rule out the impact of spatio-temporal sampling on seasonal variation patterns, we design two sensitivity cases: a "MODIS/Terra_match MISR" case in which the monthly mean AOD of MODIS/Terra is calculated using only the days when MISR overpasses, and a "MODIS/Aqua_match CALIPSO" case in which the monthly mean AOD of MODIS/Aqua is calculated using only the overpassing days of CALIPSO. The results are illustrated in Fig. 2. In all three regions, the monthly mean AODs are slightly different for "MODIS/Terra" and "MODIS/Terra_match MISR", but the seasonal variation patterns are largely the same. The same results are found for "MODIS/Aqua" and "MODIS/Aqua_match CALIPSO". As such, we conclude that sampling has little effect on the AOD seasonal variation patterns reported in this study. In fact, this conclusion is compatible with the findings of Colarco et al. (2014). Colarco et al. (2014) revealed that the spatial sampling artifacts were significant for fine aggregation grid (e.g., 0.5°), but they are reduced at coarse grid scales (e.g., 10°). In this study, we use only the mean AOD over three large regions (about 20°×20°) across 10 years, therefore the sampling artifacts are expected to be even smaller. (Line 208-224)

Line 267-269: I highly agree with this analysis. By doing this, the effects of PBL and relative humidity could be minimized.

Response: Thank you!

Line 303-304: Yang et al. (2018, AR) also showed the winter vs summer patterns (high in winter, low in summer) of inter-regional transport (between pearl river delta and Hongkong) of aerosols; Garrett et al. (2010, Tellus B) also showed the strong transport in winter and weak transport in fall for aerosols from mid-latitudes to Arctic. Actually, the vertical distribution of aerosols is very interesting in the Arctic, with high values at heights around 2-7 km, mainly caused by long-range transport from other regions such as mid-latitudes.

Response: Thank you for providing these useful references. We have included them to support the point that the seasonal variations of AOD at different vertical levels are influenced by the seasonal patterns of inter-regional transport of aerosols. (Line 328-331)

Line 374, acts -> act, strengthens -> strengthen

Response: Done. Thank you! (Line 401-403)

Line 400-405, Since AOD includes the impacts of relative humidity, is it possible that relative humidity also contributes to the diurnal variation of AOD? Another possible contribution might be the growth of fine aerosols, as indicated by Zhao et al. (2018, AAS), the growth rates of fine mode aerosols generally starts from the morning time with a growth rate of around 2-7 nm/hour (1.7-6.5 nm/hour near Beijing region in summer). They could become accumulation mode from fine mode aerosols at night time or on next day (then day time) making AOD larger.

Response: Thank you for your suggestions. We have removed the section about diurnal variations in the revised manuscript following the 2$^{nd}$ reviewer's comments. We hope this is acceptable for you.

Line 462-465: I agree that absorbing aerosols could play different roles to convection and clouds from non-absorbing aerosols. However, I think both absorbing aerosols and non-absorbing aerosols will reduce the direct solar radiation reaching the surface, causing surface cooler and further affect convection and clouds in similar way (of course, no absorption of solar radiation in the air), while the effects could be weaker than absorbing aerosols.

Response: Following the reviewer's comment, we have revised these sentences as follows:

Both absorbing and non-absorbing aerosols could invigorate deep convection by serving as cloud condensation nuclei and affect convection by reducing downward solar radiation and causing surface cooling (Rosenfeld et al., 2008). However, absorbing aerosols play unique roles in convection and cloud development by heating the atmosphere. This inhibits convection in most situations (Ramanathan et al., 2005; Massie et al., 2016; Zhao et al., 2018a) but may enhance convection and cloud formation above the PBL (Wang et al., 2013; Bond et al., 2013), depending on the vertical distribution of absorbing aerosols. (Line 445-452)

Reviewer 2:

Title: Intra-annual variations of regional aerosol optical depth, vertical distribution, and particle types from multiple satellite and ground-based observational datasets

Summary: The paper combines retrievals and observations from multiple satellites (active and passive) and ground based (in-situ and remote sensed), in order to characterize the seasonal and diurnal variations of aerosol properties in three heavily-populated regions (EUS, WEU and ECC). The aerosols are separated into lower (< 800 m) and higher levels (> 800m), monthly averages are calculated, and annual cycles plotted. Analysis and interpretation and some speculation are presented. The main conclusions are that in all three regions, column AOD and higher level AOD all peak in the summer, whereas AOD in lower levels peaks during the winter. AOD from fine-sized particles peaks in the spring/summer and is attributed to anthropogenic sources. Dust and sea-salt peaks in the winter in WEU but are nearly constant in the other two regions. There appears to be larger nighttime/daytime AOD difference in summer than winter.

Response: We thank the reviewer for the insightful comments. We have carefully addressed these comments in revising the manuscript. Point-to-point responses are given below.

This paper is logical and easy to read. The English language usage is satisfactory. The idea of separating into low level (e.g. < 800 m; presumably a proxy for boundary layer) and higher level (> 800 m) is unique. I wish I could believe all of the conclusions. But I don't yet. Like I explained in the "initial quality" review, I have strong concerns about data sampling. For example, Colarco et al., (2014, [10.5194/amt-7-2313-2014]) explains that "sampling matters", and that when we develop climatology from different types of orbits (and coverage), we need to deal with this problem. Because of this, I don't think that "the impact of the sampling issue is expected to be much smaller than that on the AOD retrieval in an individual month at a specific location" (Lines 162-167). If MODIS calculates monthly mean based on all 30 days and CALIPSO based on 2-4 times month (every 8 or 16 days, if lucky), then we don't expect the monthly means to match. Of course, if there are clouds, this could be MODIS making monthly means from, say 10 days, and CALIPSO making only one. One more paper to think about is Chin et al., (2014; [10.5194/acp-14-3657-2014]). Although they study multi-year data, they make points about comparing datasets with all kinds of sampling differences.

Response: We thank the reviewer for this valuable comment. To investigate the impact of data sampling on seasonal variation of AOD, we design two sensitivity cases: a "MODIS/Terra_match MISR" case in which the monthly mean AOD of MODIS/Terra is calculated using only the days when MISR overpasses, and a "MODIS/Aqua_match CALIPSO" case in which the monthly mean AOD of MODIS/Aqua is calculated using only the overpassing days of CALIPSO. The results are illustrated in the following figure (Fig. 2 in the revised manuscript). In all three regions, the monthly mean AODs are slightly different for "MODIS/Terra" and "MODIS/Terra_match MISR", but the seasonal variation patterns are largely the same. The same results are found for "MODIS/Aqua" and "MODIS/Aqua_match CALIPSO". As such, we conclude that sampling has little effect on the AOD seasonal variation patterns reported in this study. In fact, this conclusion is compatible with the findings of Colarco et al. (2014). Colarco et al. (2014) revealed that the spatial sampling artifacts were significant for fine aggregation grid (e.g., 0.5°), but they are reduced at coarse grid scales (e.g., 10°). In this study, we use only the mean AOD over three large regions (about 20°×20°) across 10 years, therefore the sampling artifacts are expected to be even smaller.

We have added the preceding discussions in the revised manuscript. (Line 208-224)

[Figure]

(a)  (b)  (c)

**Figure.** Monthly mean AOD observed by MISR, MODIS/Terra, MODIS/Aqua, and CALIPSO during 2007-2016 in (a) EUS, (b) WEU, and (c) ECC. For CALIPSO, only clear-sky daytime profiles are averaged in order to be consistent with the MISR and MODIS products. "MODIS/Terra_match MISR" is a sensitivity case in which the monthly mean AOD of MODIS/Terra is calculated using only the days when MISR overpasses, and "MODIS/Aqua_match

CALIPSO" is a case in which the monthly mean AOD of MODIS/Aqua is calculated using only the overpassing days of CALIPSO. The error bars denote the standard deviation of the monthly mean AOD values obtained over all years. Note the different scales on the y-axes of the plots.

Of course, the low bias (Fig 2, lines 216-218) between CALIPSO -derived AOD and the other satellites (MODIS, MISR), can be because of assumed lidar ratios (Ma et al., AMT; [10.5194/amt-6-2391-2013]), or undetected aerosol layers (Kim et al., JGR, [10.1002/2016JD025797]). I guess that the Thorson (2017) reference already listed could be a reason as well.

Response: Thank you! We have added these explanations to the revised manuscript, citing the two references. (Line 315-318)

Why the non-confident statements ("probably") in lines 227-231? I think you should be able to find references. What are the sources of these SOA? What about long-range transport in the summer?

Response: To address the reviewer's comments, we have revised these sentences as follows:

Considering the high accuracy of AERONET, we conclude that AOD peaks in summer/spring and dips in winter. An important reason for the higher AOD in summer is that the stronger radiation and higher temperature accelerate the formation of secondary aerosols (Timonen et al., 2014), including sulfate, nitrate, ammonium, and secondary organic aerosol (SOA). SOA is produced by photo-oxidation of volatile organic compounds (VOCs) and intermediate volatility organic compounds (IVOCs), as well as the chemical aging of primary organic aerosol (Zhao et al., 2016). Another reason is that more abundant water vapor in summer favors the hygroscopic growth of aerosols (Liu et al., 2012; Zheng et al., 2017). The different patterns of long range transport as a function of season is also partly responsible for the seasonable variation of AOD (Tian et al., 2017; Yang et al., 2018; Garrett et al., 2010). (Line 232-242)

Considering lines 235-236, I again ask about sampling? Are the monthly means for AERONET and satellites computed based on the same days? Or is mean AERONET = mean (of AERONET data) and mean satellite = mean (of SATELLITE data)? We know from validation exercises that when actually collocated in space and time (both AERONET and satellite are free of clouds) that they match overall well (yes, sometimes small biases, e.g. Remer et al., 2005). However, I do not expect matches if using different samples (days). Note that the Remer et al., (2005) study has been updated for MODIS (Collection 5 and Collection 6), and there are also updates for MISR evaluations. The "instrument calibration issue" (lines 243-244) would not cause such a large bias.

Response: For each satellite-borne sensor, only those days for which the satellite overpasses an AERONENT site were used in the comparisons. In other words, the monthly means for AERONET and satellites were indeed computed based on the same days. In addition, to match coincident measurements, the AERONET AOD retrievals for each site were averaged within a 2 h window centered on the satellite overpass times (about 10:30 for MISR and MODIS/Terra, and 13:30 for MODIS/Aqua and CALIPSO, depending on site location), and compared with the satellite AOD retrievals in a $1° \times 1°$ grid box that contains the corresponding AERONET site. (Line 172-177)

Thank you for pointing out that the Remer et al. (2005) study is outdated. For MISR, however, the Kahn et al. (2010) study is applicable to the product used in this paper. We have revised the descriptions about the discrepancies among MODIS, MISR and AERONET as follows:

While relative patterns of AOD seasonal variations from observations of MISR, MODIS/Terra, and MODIS/Aqua are similar to each other and to those of AERONET, the magnitude of AOD observed by these sensors shows considerable discrepancies. In all three regions, the AOD retrieved from MODIS is larger than that from MISR, consistent with the results of previous studies (de Meij et al., 2012; Zhao et al., 2017; Chin et al., 2014; Kang et al., 2016; Qi et al., 2013). This is most likely due to differences in observing strategy, retrieval algorithms, and spatio-temporal sampling (Kahn et al., 2009). The MISR-retrieved AOD agrees well with the AERONET observations in EUS and WEU regions. In the ECC region, however, MISR underestimates the AERONET AOD, probably because there is less signal from the surface at higher AOD, which creates ambiguity that can result in the algorithm assigning too much of the top-of-atmosphere radiance to the surface (i.e., a higher surface albedo), thereby underestimating the AOD (Kahn et al., 2010). The MODIS/Terra and MODIS/Aqua overestimate the AERONET AOD to some extent in all three regions. The overestimation was also reported in two previous studies (de Meij et al., 2012; Ruiz-Arias et al., 2013) using the level 3 MODIS products (Collection 5 or 5.1). We show a relatively larger overestimation than that reported by de Meij et al. (2012) and Ruiz-Arias et al. (2013), partly because we used the AERONET AOD averaged within a 2 h window centered on the satellite overpass times while the two previous studies used the daily/monthly mean AERONET AOD in the comparisons. The daily mean AOD observed by AERONET is about 10% larger than the value during the satellite overpass times (Li et al., 2013). The reasons for the overestimation are yet to be thoroughly elucidated in future studies. (Line 243-262)

I think it is a good idea that you are comparing low-level CALIPSO to ground level PM2.5 (lines 267-269) but I wonder about the temporal sampling. Also, PM2.5 is usually a "dried" aerosol measurement whereas CALIPSO is ambient RH.

Response: In the original manuscript, the monthly mean $PM_{2.5}$ concentrations were calculated using observations in all days. Here we redo the calculation using only the days when CALIPSO overpasses an observational site (dashed lines in the following figure, shown below), and compare with the original estimates (solid lines). The results show that the temporal sampling has minor effects on the monthly mean $PM_{2.5}$ concentrations. In the revised manuscript, we used the updated calculation method (dashed lines) in order to match the CALIPSO observations.

[Figure]

**Figure.** Monthly mean surface $PM_{2.5}$ concentrations during 2007-2016 in three target regions. The solid lines represent monthly mean $PM_{2.5}$ concentrations calculated using observations in all days, while the dashed lines are calculated using only the days when CALIPSO overpasses an observational site. The numbers of observational sites included in averaging are 225, 52, and 496, in the EUS, WEU, and ECC regions. Note the different scales on the y-axes for EUS/WEU and ECC.

We agree with the reviewer that the low-level AOD observed by CALIPSO is affected by ambient RH. Nevertheless, previous studies have reported fairly good correlations between extinction coefficient/low-level AOD and $PM_{2.5}$ concentrations (Cheng et al., 2013; Zheng et al., 2017). In addition, we intend to do a qualitative and not quantitative comparison. For these reasons, it appears reasonable to compare the seasonal variation patterns of low-level AOD and $PM_{2.5}$ concentrations. We have included the discussions in the revised manuscript (Line 281-284).

I don't understand the arguments in lines 279-282, in that since CALIPSO can't detect thin aerosols, that the fraction of upper-level aerosols is smaller than at the surface, and that results in the CALIPSO AOD as being weighted toward lower heights. According to the Kim et al., paper (listed above), CALIPSO is likely to miss stuff close to the ground. Anyway, the point is I don't think you can say that CALIPSO is missing stuff, and yet it "provides valuable information with respect to intra-annual variations at specific height ranges" (line 289-290).

Response: Indeed, the aerosols with heights below 200 m AGL are frequently undetected because of surface contamination (Kim et al., 2017; NASA CALIPSO team, 2011), but the overall fraction of aerosols detected in the upper levels (> 800 m AGL) is still much smaller than that in the lower levels (< 800 m AGL) because the upper-level aerosols tend to be optically thin. This is evident from Fig. 10 of Kim et al. (2017) and Fig. 1 of Thorsen et al. (2017). Therefore, the CALIPSO-observed AOD seasonal variations are significantly weighted toward lower heights.

The detection sensitivities in the upper and lower levels differ significantly because the extinction coefficient decreases by about 2 orders of magnitude with an increase of height (Kim et al., 2017; Thorsen et al., 2017). Within a specific height range, however, the optical thickness of aerosols and hence the detection fraction has a smaller variability. This is supported by the fact that the seasonal mean AOD within a specific height range differs by at most 3 times as a function of season (Fig. 4 in the main text). For these reasons, we argue that CALIPSO could provide valuable information with respect to seasonal variations of aerosols within a specific height range.

We have added the preceding discussions in the revised manuscript. (Line 295-304, 309-315)

Thank you for adding many references in lines 292-304 to discuss why AOD seasonal differences should be different in lower versus higher altitudes. I don't know if I agree that "seasonal variations of AOD at different levels are influenced by variations of RH which affects hygroscopic growth" (Lines 301-303). Of course, RH influences AOD, but it is total column water vapor and not necessarily RH that changes drastically from season to season.

Response: Following the reviewer's comment, we have changed "RH" to "water vapor amount".

I agree that comparing MISR-derived aerosol "types (size/ absorption)" and CALIPSO derived aerosol "types" (sources) is ultimately useful. (lines 307-308). However, they are clearly different beasts, and I am getting lost reading this section (Section 3.3). Each paragraph has multiple sentences that are "A implies B, whereas (while / in contrast) sometimes C implies D". It's hard to follow. I suggest a table, or schematic cartoons, or bullets.

Response: Following the reviewer's comment, we have added a table summarizing the seasonal variations of different aerosol types in the three regions (shown below). We have also carefully revised the text of this section improve the logic and readability. (Line 333-408)

**Table 1.** Summary of the seasonal variations of the total, height-specific, and type-specific AOD

|  | EUS | WEU | ECC |
|---|---|---|---|
| Total column AOD | Peak in summer | Peak in summer/late spring | Peak in summer/spring |
| AOD > 800 m AGL | Peak in summer | Peak in summer/late spring | Peak in summer/spring |
| AOD < 800 m AGL | Two peaks in winter and summer | Peak in winter | Peak in winter |
| Small-size | Peak in summer | Peak in summer/late spring | Peak in summer/spring |
| Medium-size | Peak in summer | Peak in summer/late spring | Peak in summer/spring |
| Large-size | Rather uniform | Rather uniform | Peak in spring |
| Absorbing | Peak in summer | Peak in summer/late spring | Two peaks in Mar and Aug |
| Polluted continental/dust | Similar to height-specific total AOD | Similar to height-specific total AOD | Similar to height-specific total AOD |
| Dust | No obvious seasonal pattern | Peak in summer | Peak in spring |
| Clean marine | No obvious seasonal pattern | Peak in winter | Negligible amount |
| Smoke | Peak in summer | Peak in summer/late spring | Two peaks in Mar and Aug |

I notice that regarding the aerosol typing as seen by CALIPSO, all of the regions (over land), have non-trivial amount of "clean-marine" aerosol (Fig 7). Is this transported marine aerosol to the entirety of the regional box, or should the marine aerosol be expected to be more dominant but confined only to the coastal areas of a region?

Response: All three regions used in the study cover some ocean areas (see Fig. 1 in the manuscript). The marine aerosols are predominantly located over the ocean and in coastal areas, and are much fewer over land.

The section on daytime/nighttime variability is nice, but I think it is beside the point of the rest of the paper. Why would smoke AOD accumulate at night? Higher RH at night might make bigger aerosols, but if anything, fire activity is reduced at night. You might check the PM2.5 measurements here. I suggest leaving this section out, and thinking about the questions related to the other sections. "Intra-annual", "vertical", and "particle types" is enough for one paper!

Response: Following the reviewer's suggestion, we have left out this section in the revised manuscript.

In terms of figures. I can see why the authors do this (different magnitudes of AOD and or PM2.5 at different sites), but the varying y-axes within figure captions, and from figure-to-figure are distracting. But thank you for pointing out in the caption!

Response: We tried to unify the scales of the y-axes but failed because the magnitude differs greatly according to figures. Thank you for your understanding.

What is "upper air"? I see it a few places, and assume you mean > 800 m AGL? (e.g. Line 300).

Response: Yes, it means > 800 m AGL. We have explained this in the revised manuscript. (Line 299)

The abstract suggests (lines 37-38) that results can "help to improve the current estimates of climatic and health impacts of aerosols". Well maybe, but I would drop this from the abstract since there is no discussion in the paper.

Response: Following the reviewer's comment, we have removed this sentence from the abstract.

**References**

[revised manuscript text omitted]